# High Prevalence of Atrial Fibrillation in a Lithuanian Stroke Patient Cohort

**DOI:** 10.3390/medicina58060800

**Published:** 2022-06-14

**Authors:** Rytis Masiliūnas, Austėja Dapkutė, Julija Grigaitė, Jokūbas Lapė, Domantas Valančius, Justinas Bacevičius, Rimgaudas Katkus, Aleksandras Vilionskis, Aušra Klimašauskienė, Aleksandra Ekkert, Dalius Jatužis

**Affiliations:** 1Center of Neurology, Vilnius University, 08661 Vilnius, Lithuania; austeja.dapkute@santa.lt (A.D.); julija.grigaite@santa.lt (J.G.); domantas.valancius@santa.lt (D.V.); ausra.klimasauskiene@santa.lt (A.K.); aleksandra.ekkert@santa.lt (A.E.); dalius.jatuzis@santa.lt (D.J.); 2Faculty of Medicine, Vilnius University, 03101 Vilnius, Lithuania; jokubaslape@gmail.com; 3Center of Cardiology and Angiology, Vilnius University, 08661 Vilnius, Lithuania; justinas.bacevicius@santa.lt (J.B.); rimgaudas.katkus@santa.lt (R.K.); 4Clinic of Neurology and Neurosurgery, Vilnius University, 03101 Vilnius, Lithuania; aleksandras.vilionskis@rvul.lt

**Keywords:** ischemic stroke, atrial fibrillation, antithrombotic treatment, Lithuania, survival

## Abstract

*Background and Objectives*: Atrial fibrillation (AF) is the most common cardiac arrhythmia and is associated with a five-fold increased risk for acute ischemic stroke (AIS). We aimed to estimate the prevalence of AF in a Lithuanian cohort of stroke patients, and its impact on patients regarding case fatality, functional outcome, and health-related quality of life (HRQoL) at 90 days. *Materials and Methods*: A single-center prospective study was carried out for four non-consecutive months between December 2018 and July 2019 in one of the two comprehensive stroke centers in Eastern Lithuania. A telephone-based follow-up was conveyed at 90 days using the modified Rankin Scale (mRS) and EuroQoL five-dimensional three-level descriptive system (EQ-5D-3L) with a self-rated visual analog scale (EQ-VAS). One-year case fatality was investigated. *Results*: We included 238 AIS patients with a mean age of 71.4 ± 11.9 years of whom 45.0% were female. A striking 97 (40.8%) AIS patients had a concomitant AF, in 68 (70.1%) of whom the AF was pre-existing. The AIS patients with AF were at a significantly higher risk for a large vessel occlusion (LVO; odds ratio 2.72 [95% CI 1.38–5.49], *p* = 0.004), and had a more severe neurological impairment at presentation (median NIHSS score (interquartile range): 9 (6–16) vs. 6 (3–9), *p* < 0.001). The LVO status was only detected in those who had received computed tomography angiography. Fifty-five (80.9%) patients with pre-existing AF received insufficient anticoagulation at stroke onset. All patients received a 12-lead ECG, however, in-hospital 24-h Holter monitoring was only performed in 3.4% of AIS patients without pre-existing AF. Although multivariate analyses found no statistically significant difference in one-year stroke patient survival and favorable functional status (mRS 0–2) at 90 days, when adjusted for age, gender, reperfusion treatment, baseline functional status, and baseline NIHSS, stroke patients with AF had a significantly poorer self-perceived HRQoL, indicated by a lower EQ-VAS score (regression coefficient ± standard error: β = −11.776 ± 4.850, *p* = 0.017). *Conclusions*: In our single-center prospective observational study in Lithuania, we found that 40.8% of AIS patients had a concomitant AF, were at a higher risk for an LVO, and had a significantly poorer self-perceived HRQoL at 90 days. Despite the high AF prevalence, diagnostic tools for subclinical AF were greatly underutilized.

## 1. Introduction

Atrial fibrillation (AF) is the most common cardiac arrhythmia, affecting 33.5 million people globally, and is a well-established independent risk factor for acute ischemic stroke (AIS) [1,2]. Cardioembolic stroke is known to be more severe [3] and associated with greater morbidity, mortality, and disability [4,5]. In addition, stroke patients with AF exhibit a high risk of recurrent ischemic events [6,7]. Therefore, detecting AF is crucial both for primary and secondary stroke prevention. The earlier and the longer cardiac monitoring occurs after a stroke, the higher the chance to detect AF and consequently prescribe the right antithrombotic treatment for secondary prevention of stroke [8,9].

Oral anticoagulation (OAC) reduces the risk of stroke and all-cause mortality compared with control or placebo in patients with non-valvular AF [10,11]. In addition, there is no evidence of better stroke outcomes in patients with AF taking antiplatelet agents versus not taking any antithrombotic medication [12,13]. However, OAC is frequently under-prescribed, especially in the elderly [14], and of those treated, the proportion of adherent patients could be as low as 41% one year later [15,16,17].

The incidence of AIS and stroke mortality in Lithuania is amongst the highest in the world [18,19], in large part due to a high prevalence and poor control of cardiovascular risk factors [18,20]. Moreover, in Lithuania the expected increase in age-adjusted stroke incidence and prevalence rates is the highest, and the expected decrease in stroke mortality is the lowest among all European Union countries [21]. However, data regarding the proportion of AIS cases associated with AF is lacking, although studies from the other two Baltic States indicate some of the highest AF prevalence among AIS patients globally, reaching up to a striking 48.6% [22,23,24].

The aim of our study was (1) to estimate the prevalence of AF in a Lithuanian cohort of stroke patients, and its (2) impact on patients’ case fatality, functional outcome, and health-related quality of life (HRQoL) at 90 days.

## 2. Materials and Methods

### 2.1. Study Design and Population

A single-center prospective study was carried out for four full non-consecutive months to account for seasonal differences—between December 2018 and July 2019. All patients with AIS, treated in Vilnius University Hospital—one of the two comprehensive stroke centers in Eastern Lithuania—were included.

The study was approved by the Lithuanian Bioethics Committee and complied with STROBE guidelines for observational research.

### 2.2. Baseline Characteristics

Demographic and clinical characteristics such as age, sex, presence of AF, National Institutes of Health Stroke Scale (NIHSS) score on admission, presence of a large vessel occlusion (LVO), reperfusion therapy, antithrombotic treatment used, and AF screening status were collected. The AF screening included 24-h Holter monitoring only, as Stroke Unit telemetry was not routinely performed during the study period. Pre-existing AF was defined as AF known prior to a stroke onset. Atrial fibrillation detected after a stroke (AFDAS) was defined as a newly detected AF after a stroke in patients without pre-existing AF [25].

### 2.3. Follow-Up and Outcomes

The main outcomes for this analysis were: (1) all-cause in patient, 90-day, and 1-year case fatality, (2) favorable functional outcome at 90 days (modified Rankin Scale (mRS) 0–2), and (3) self-reported HRQoL at 90 days, measured by the EuroQoL five-dimensional three-level descriptive system (EQ-5D-3L).

Either AIS patients or their caregivers who could participate in a telephone-based follow-up were surveyed 90 days after stroke onset using the mRS. In addition, the EQ-5D-3L with a self-rated visual analog scale (EQ-VAS) was used as a self-reported measure of HRQoL. The EQ-5D-3L is a widely used instrument, describing the respondent’s health state in terms of three severity levels in each of the five domains: mobility, self-care, usual activities, pain/discomfort, and anxiety/depression [26,27]. Individual values were transformed into an index value ranging from −0.074 to 1 using a European value set with 1 being the best health possible, 0 being dead, and a score < 0 representing a health condition worse than death [28]. The EQ-VAS provided information about the participants’ subjective health perception: the AIS patients were asked to score their health state on that specific day of the survey on a scale from 0 to 100, with 0 being “the worst health you can imagine” and 100 being “the best health you can imagine”.

Information on all-cause patient case fatality, including the date of death, was obtained retrospectively from electronic health records one year after stroke onset. It is mandatory to issue all death certificates electronically in Lithuania [29], therefore, precise information on all patients was available.

### 2.4. Statistical Analysis

The qualitative variables were presented as count and percentage. The quantitative variables, based on their Gaussian distribution, were presented as mean ± standard deviation (SD) or median and interquartile range (IQR). Individuals were categorized into groups by the presence of AF. The AF group was further divided depending on whether the AF was known prior to admission or was detected after the stroke during the hospital stay. Student’s *t*-test or Mann–Whitney U test was used to compare quantitative variables, as appropriate. For categorical variables, Chi-square test or Fisher’s exact test was used, and 95% confidence intervals (CI) were calculated, as appropriate. We used a binary multiple logistic regression model to calculate the odds ratio (OR) of having an LVO in patients with pre-existing or newly diagnosed AF, adjusted by age and sex. Our definition of an LVO included the occlusion sites accessible for thrombectomy: middle cerebral artery 1st and 2nd segments (M1, M2), anterior cerebral artery (A1 segment), posterior cerebral artery 1st and 2nd segments (P1, P2), basilar artery, vertebral artery, and internal carotid artery [30].

“Time 0” for survival analyses was the date of hospitalization for the index stroke event. A comparison of the probability of freedom from the prognostic binary endpoints between groups with and without AF was performed by Kaplan–Meier survival analysis. Differences in the Kaplan–Meier curves were evaluated with the log-rank test. Univariate and multivariate Cox regression analyses were also performed to determine the prognostic implications of each variable on patient death. Variables investigated included AF, age, gender, presence of reperfusion treatment, good baseline functional status (mRS 0–2), and baseline NIHSS. Variables with *p* < 0.1 on univariate analysis were entered into the multivariable model. The hazard ratios (HR) were presented with their 95% confidence intervals.

In addition, both unadjusted and adjusted logistic regression analyses were performed with a good functional outcome (mRS 0–2) as the dependent variable. The adjusted logistic analyses were performed using stepwise forward selection, based on significant variables in the adjusted models.

Finally, multiple linear regression was used to estimate the associations between the independent variables (age, gender, AF, reperfusion treatment, good baseline functional status, and baseline NIHSS) and HRQoL, represented by EQ-VAS and EQ-5D-3L index scores. A value of *p* < 0.05 (two-sided) was considered to be statistically significant. The software R version 3.6.2 (The R Foundation for Statistical Computing, Vienna, Austria) was used for all statistical analyses.

## 3. Results

### 3.1. Demographic and Clinical Characteristics

Overall, 238 AIS patients with a mean age of 71.4 ± 11.9 years were included in our study of which 45.0% were female (Table 1). Ninety-seven (40.8%) AIS patients were found to have concomitant atrial fibrillation: 68 (28.6%) were diagnosed with AF prior to hospital arrival, and 29 (12.2%) had AFDAS (Table 2, Figure 1, Appendix A). While comparing the groups with and without AF, stroke patients with AF were significantly older (75.7 vs. 68.4 years, *p* < 0.001), and had a more severe neurological impairment at presentation (a median NIHSS score of 9 [IQR 6–16] vs. 6 [3,4,5,6,7,8,9], *p* < 0.001). However, the baseline level of functional independence (mRS 0–2) did not differ between groups and reached an overall 89.5%. In addition, the proportion of female patients was significantly larger in the AF group (58.8% vs. 35.5%, *p* < 0.001).

A total of 160 (67.2%) AIS patients underwent computed tomography angiography (CTA), in 51.3% of whom an LVO was found. Patients with AF were at a significantly higher risk for an LVO compared to patients without AF with an odds ratio of 2.72 (95% CI 1.38–5.49, *p* = 0.004), when adjusted by age and gender. Sixty-one (38.1%) patients had an LVO in the anterior circulation, 17 (10.6%) in the posterior circulation, and 4 (2.5%) patients had an LVO in both. Anterior circulation stroke was significantly more common in patients with a known LVO and AF (55.2% vs. 25.8%, *p* < 0.001).

### 3.2. Antithrombotic Treatment

Out of 68 patients with pre-existing AF, 11 (16.2%) took non-vitamin K antagonist oral anticoagulants (NOACs), 26 (36.8%) vitamin K antagonists, 8 (11.8%) antiplatelets, and 23 (33.8%) did not take any antithrombotic treatment at index stroke event (Table 2).

Out of 26 (36.8%) patients on vitamin K antagonists, only five (19.2%) had an International Normalized Ratio (INR) within the therapeutic range on admission. Strikingly, 55 (80.9%) stroke patients with pre-existing AF at stroke onset were receiving insufficient anticoagulation (received no antithrombotic treatment, took antiplatelets, warfarin with INR < 2, or had inadequate NOAC dosing). The therapeutic range for INR was considered to be 2.0–3.0.

At discharge, 64 (76.2%) patients were prescribed oral anticoagulants—23 (27.4%) warfarin and 41 (48.8%) NOACs. Out of the 20 remaining patients, 10 (11.9%) were given low-molecular-weight heparins and 10 (11.9%) were prescribed treatment with antiplatelets. In the antiplatelets group, oral anticoagulants were contraindicated for four patients due to a high risk of bleeding, for one patient due to high stroke severity, and five (6.0%) patients with AF had no documented contraindications (unexplained reasons).

### 3.3. Screening for AF

All patients received at least one 12-lead ECG, however, only five (3.4%) patients without pre-existing AF had 24-Holter monitoring performed during the hospitalization for the index event.

### 3.4. Patient Survival, Functional Outcome, and HRQoL

The Kaplan–Meier survival curve displays the 1-year probability of survival of AIS patients with and without AF (Figure 2). The overall all-cause case fatality was greatest in the first 90 days after stroke (21.8%; 95% CI: 16.8–27.6%), and the 1-year cumulative risk of death was 29.0% (95% CI: 23.3–35.2%) (Table 3). A log-rank test for a difference in time to death among stroke patients with and without AF was significant (*p* < 0.001), in favor of patients without AF. In univariate Cox regression analysis, age (*p* < 0.001), presence of AF (*p* < 0.001), baseline good functional status (*p* < 0.001), and baseline NIHSS (*p* < 0.001) were found to be significant factors that have influence on overall stroke patient survival (Table 4). However, in the multivariable model, only age (*p* = 0.007), and baseline NIHSS (*p* < 0.001) contributed significantly.

A total of 133 out of 186 surviving patients or caregivers completed the mRS survey at 90 days; 58.6% of respondents had an mRS score ≤ 2 (56.3% in stroke with AF group, 60.0% in stroke without AF group, *p* = 0.673), and were considered to be able to look after themselves without daily assistance (Table 3). However, AF did not have a statistically significant influence on the favorable functional outcome (mRS 0–2) in a multivariable binary logistic regression analysis, where the presence of reperfusion treatment (OR 3.91 [95% CI 1.66–10.05], *p* = 0.003) and the baseline NIHSS score (OR 0.82 [95% CI 0.73–0.90], *p* < 0.001) were the only two significant factors (Appendix A).

A total of 126 patients or caregivers completed the EQ-5D-3L survey and 111 AIS patients evaluated their EQ-VAS score 90 days after the index AIS event. Stroke patients with AF evaluated their EQ-VAS score significantly worse in comparison to those without AF (50 [IQR: 30–70] vs. 60 [40–76.25], *p* = 0.022), suggesting a lower perception of the HRQoL (Table 3). However, the EQ-5D-3L index scores did not differ significantly between groups. Although multivariate linear regression analyses showed that AF had a significant influence on EQ-VAS (β ± SE: −11.776 ± 4.850, *p* = 0.017), no such influence was demonstrated for the EQ-5D-3L index score (β ± SE: −0.013 ± 0.060, *p* = 0.833), when adjusted for age, gender, presence of reperfusion treatment, baseline good functional status, and baseline NIHSS score (Appendix A). In addition, the only statistically significant difference between the five EQ-5D-3L domains’ values was found for difficulty with self-care, which was more common in patients with AF (48.9% vs. 36.9, *p* = 0.041) (Table 3).

### 3.5. Stroke Patients with AFDAS

A history of stroke/TIA and congestive heart failure were significantly less common in AIS patients with AFDAS as compared to AIS patients with pre-existing AF (10.3% vs. 32.4, *p* = 0.024, and 44.8% vs. 66.2, *p* = 0.050, respectively). In addition, stroke patients with AFDAS evaluated their EQ-VAS score as significantly better in comparison to those with pre-existing AF, even when adjusted for age, gender, cardiovascular risk factors, reperfusion treatment, baseline functional status, and baseline NIHSS (42.5 [IQR: 22.5–57.5] vs. 70 [50–75], *p* = 0.022) (Appendix A).

## 4. Discussion

In our single-center prospective observational study, we found that 40.8% of AIS patients had a concomitant AF—one of the largest figures in the available literature. In addition, 80.9% of stroke patients with pre-existing AF received insufficient anticoagulation at stroke onset. Finally, although multivariate analyses found no statistically significant difference in one-year stroke patient survival and functional status at 90 days, when adjusted for age, gender, reperfusion treatment, baseline functional status, and baseline NIHSS, stroke patients with AF had a significantly worse self-perceived HRQoL, indicated by a lower EQ-VAS score.

The prevalence of AF among AIS patients varies between countries and was found to be as high as 31–38% in a population-based registry in Greece [31,32], and up to 48.6% in two separate hospital-based registries in neighboring Latvia [22,24]. The high prevalence of AF found in our investigated AIS patient population could indicate a regional trend, as other cardiovascular risk factors are largely similar within the Baltic States [32]. This contrasts with the fact that the regional prevalence of AF in the Eastern European general population is lower than in Western Europe and North America [2]. Herein, a small penetration of NOACs due to the poor reimbursement conditions could play a role, as NOACs are reimbursed only following the warfarin failure or contraindications, and cannot be initially prescribed by a general practitioner or a neurologist—a policy previously common in the Central and Eastern European states [33]. This reasoning is supported in our patient cohort by the low rate of OAC in patients with pre-existing AF before stroke onset. Although a patient’s personal decision to deliberately discontinue OAC in asymptomatic AF or currently normal rhythm could also play a role.

Our sample confirmed the association between AF in AIS patients and an LVO—a fact long identified in previous literature [34]. While LVOs are present in less than one-third of stroke cases, they contribute to around three-fifths of post-stroke dependence and death, and more than 90% of post-stroke mortality [35]. Therefore, it is crucial to ensure adequate AF screening in AIS patients, as prompt administration of OAC could contribute to an 8.4% annual absolute risk reduction of stroke recurrence compared with antiplatelet therapy [36].

The 2018 American Heart Association/American Stroke Association guidelines state that cardiac monitoring for potential arrhythmia should be performed for at least the first 24 h after a stroke [37]. Whereas the 2020 European Society of Cardiology guidelines recommend extending cardiac monitoring to at least 72 h in patients after AIS without pre-existing AF [38]. Finally, in patients with ischemic stroke of undetermined origin, the use of implantable devices for long-term cardiac monitoring instead of non-implantable devices is recommended by the recent European Stroke Organization guidelines [38,39]. However, we found that a mere 3.4% of the AIS patients without pre-existing AF underwent 24-h Holter monitoring while in hospital, suggesting that AF prevalence could be even higher. This may be due to the poor availability of 24-h Holter devices in addition to the lack of strict standard operational procedures, as the current national Lithuanian stroke care guidelines do not mention extensive cardiac monitoring [40].

There is a growing trend in utilizing new wearable devices for screening for AF or even mitigating the risks of patients with known AF, guided by the newest consensus document of the European Heart Rhythm Association [41]. This is especially so as previous randomized controlled trials have demonstrated that a composite outcome of ‘ischemic stroke/systemic thromboembolism, death, and rehospitalization’ could be lower with mobile AF application interventions compared with usual care [42]. However, more randomized controlled trials are needed to prove the concept in a hospital-based AIS setting.

Although multivariate analysis found no statistically significant difference in one-year stroke patient survival and functional status at 90 days, multiple previous studies have shown that AF could be associated with an increased risk of death and severe disability after AIS [4,5,43]. As the magnitude of the association is shown to be substantially diminished after multivariable adjustment, much of the association can be explained by other factors. Our findings are in line with other studies, showing that the most important determinants underlying the association between AF and mortality after ischemic stroke are age and baseline stroke severity [43]. We speculate that the small sample size precluded us from finding a significant association, and further studies with a larger study population should give more robust conclusions.

In our study population, patients with previously diagnosed AF were found to have a significantly lower perception of their HRQoL—a finding reiterated from previous studies [44]. However, a significant difference was observed only in EQ-VAS, but not in EQ-5D-3L index scores—a pattern that had been observed previously [45]. Furthermore, stroke patients with pre-existing AF evaluated their EQ-VAS score as significantly worse in comparison to those with AFDAS, indicating that a subgroup of patients with known AF had the worst self-perceived quality of life at three months. This is consistent with previous literature in which pre-existing AF has been shown to be associated with a higher disease burden and worse clinical outcomes [25]. Nevertheless, these patterns must be interpreted cautiously due to a small patient sample.

Our study is the first on the Lithuanian population that estimates the prevalence of AF among a hospital-based AIS patient cohort, and investigates its impact on patients’ case fatality, HRQoL, and long-term functional outcomes. In addition, it raises awareness about the potential need for more stringent national guidelines for cardiac monitoring in stroke patients, as suggested by the exceptionally low in-hospital 24-h Holter monitoring rates in the studied population. Finally, we explore the profile of antithrombotic treatment received by AIS patients with AF, discussing the possibility of insufficient anticoagulation, as it may be one of the overlooked causes of stroke in this group of patients.

Our study has several limitations. Foremost, as this was a single-center study with a rather small sample size, our findings could have limited generalizability, and some weak statistical associations could have been missed. Furthermore, due to the small rate of in-hospital 24-h Holter monitoring, some patients could have been misclassified as not having AF, which could have impacted the overall differences in case fatality and functional outcome. In addition, the use of antithrombotic medication was determined from documentation only, and anti-factor Xa activity was not measured, amounting to some error in OAC adherence. Fourth, the 90-day follow-up for assessing the functional status and HRQoL was telephone-based, and in many cases, caregivers responded to the survey, which makes it difficult to compare to studies where HRQoL is self-reported using visual aids. Fifth, we used the EQ-5D-3L which has been validated for stroke patients [46], but as there is no Lithuanian value set available to this date, a European value set was used. Finally, a considerable proportion of the patients were lost to the 90-day follow-up survey due to a short period of time when the investigators could not conduct the telephone survey because of technical reasons. However, because there was no bias towards a certain group of patients, we believe that this should not have affected our results.

## 5. Conclusions

In our single-center prospective observational study in Lithuania, we found that 40.8% of AIS patients had a concomitant AF, were at a higher risk for an LVO, and had a significantly poorer self-perceived HRQoL at 90 days. Despite the high AF prevalence, only a small proportion of subjects received proper anticoagulation at stroke onset, whereas in-hospital diagnostic tools for subclinical AF were greatly underutilized.

## Figures and Tables

**Figure 1 medicina-58-00800-f001:**
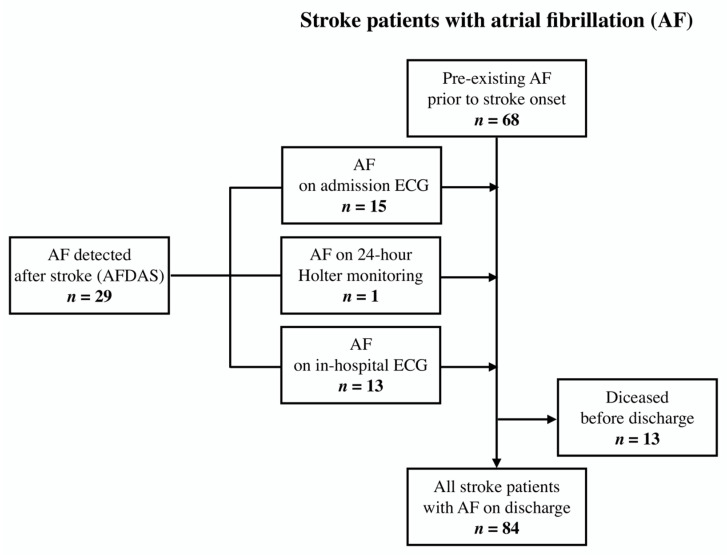
Flowchart of acute ischemic stroke patients with atrial fibrillation.

**Figure 2 medicina-58-00800-f002:**
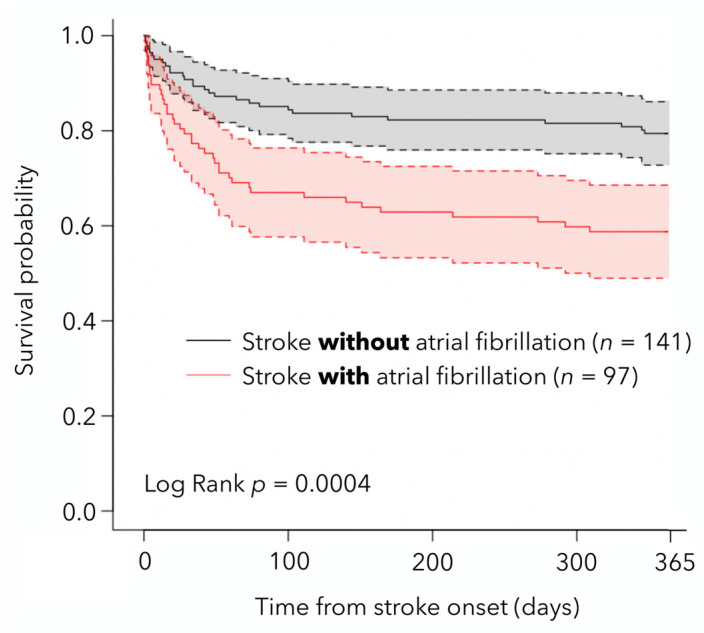
Kaplan–Meier survival plots and 95% confidence intervals for ischemic stroke patients with and without atrial fibrillation.

**Table 1 medicina-58-00800-t001:** Demographic and clinical characteristics for stroke patients with and without atrial fibrillation. Significant *p*-values are shown in bold.

	All Stroke Patients (*n* = 238)		Stroke Patients with AF (*n* = 97)		Stroke Patients without AF (*n* = 141)		*p*-Value
Female, *n* (%)	107	(45.0)	57	(58.8)	50	(35.5)	<0.001
Mean age, years (SD)	71.4	(11.9)	75.7	(11.0)	68.4	(11.5)	<0.001
Baseline median mRS ≤ 2, *n* (%)	213	(89.5)	88	(90.7)	125	(88.7)	0.609
Baseline NIHSS, median (IQR)	6	(4–12)	9	(6–16)	6	(3–9)	<0.001
Risk factors, *n* (%)							
Hypertension	217	(91.2)	93	(95.9)	124	(87.9)	0.034
Diabetes mellitus	50	(21.0)	26	(26.8)	24	(17.0)	0.069
Dyslipidemia	185	(77.7)	65	(67.0)	120	(84.1)	<0.001
History of stroke/TIA	51	(21.4)	25	(25.8)	26	(18.4)	0.175
Congestive heart failure	91	(38.2)	58	(59.8)	33	(23.4)	<0.001
Coronary artery disease	82	(34.5)	51	(52.6)	31	(22.0)	<0.001
Peripheral artery disease	12	(5.0)	5	(5.2)	7	(5.0)	0.947
Underlying malignancy	15	(6.3)	4	(4.1)	11	(7.8)	0.251
CTA performed, *n* (%)	160	(67.2)	67	(69.1)	93	(66.0)	0.615
Reperfusion, *n* (%)	91	(38.2)	38	(39.2)	53	(37.6)	
Not eligible	147	(61.8)	59	(60.8)	88	(62.4)	0.805
IVT	41	(17.2)	13	(13.4)	28	(19.9)	0.195
EVT	41	(17.2)	20	(20.6)	21	(14.9)	0.250
Combined treatment	9	(3.8)	5	(5.2)	4	(2.8)	0.357
Large vessel occlusion, *n* (%) †	82	(51.3)	44	(65.7)	38	(40.9)	0.002
Anterior	61	(38.1)	37	(55.2)	24	(25.8)	<0.001
Posterior	17	(10.6)	5	(7.5)	12	(12.9)	0.271
Both	4	(2.5)	2	(3.2)	2	(2.0)	0.739
None	78	(48.8)	23	(34.3)	55	(59.1)	0.002
24-h Holter monitoring, *n* (%) ‡							
Performed					5	(3.5)	
Not performed					136	(96.5)	

AF—atrial fibrillation, SD—standard deviation, mRS—modified Rankin Scale, NIHSS—National Institutes of Health Stroke Scale, IQR—interquartile range, TIA—transient ischemic attack, CTA—computed tomography angiography, IVT—intravenous thrombolysis, EVT—endovascular treatment. † Out of those in whom CTA was performed. ‡ Only stroke patients without known AF.

**Table 2 medicina-58-00800-t002:** Antithrombotic treatment status for stroke patients with pre-existing atrial fibrillation (AF) on admission and for all AF patients on discharge.

	Pre-Existing AF before Stroke Onset (*n* = 68)		All AF on Discharge ¶ (*n* = 84)	
Antithrombotic treatment status, *n* (%)				
No treatment	23	(33.8)	0	(0)
Antiplatelets	8	(11.8)	10	(11.9)
LMWH †	0	(0)	10	(11.9)
Warfarin †	26	(36.8)	23	(27.4)
NOACs †	11	(16.2)	41	(48.8)
INR, median (IQR) ‡	1.28	(1.20–1.79)		
INR within therapeutic range, *n* (%) ‡	5	(19.2)		
Insufficient anticoagulation, *n* (%) §	55	(80.9)		

AF—atrial fibrillation, LMWH—low molecular weight heparin, NOACs—non-vitamin K antagonist oral anticoagulants, INR—international normalized ratio, IQR—interquartile range. † With or without antiplatelets. ‡ Only for patients using warfarin (*n* = 26). § Includes AF patients with no treatment, treated with antiplatelets, INR < 2, and inadequate NOAC dosing. ¶ Excluding diseased patients.

**Table 3 medicina-58-00800-t003:** Case fatality, functional outcome, and health-related quality of life of stroke patients with and without atrial fibrillation at 90 days.

	All Stroke Patients (*n* = 238)		Stroke Patients with AF (*n* = 97)		Stroke Patients without AF (n = 141)		*p*-Value
Case fatality, *n* (%)							
In-hospital case fatality	19	(8.0)	13	(13.4)	6	4.3)	0.011
90-day case fatality	52	(21.8)	31	(32.0)	21	(14.9)	0.002
1-year case fatality	69	(29.0)	40	(41.2)	29	(20.6)	<0.001
Median mRS ≤ 2 at 90 days, *n* (%) †	78	(58.6)	27	(56.3)	51	(60.0)	0.673
Missing mRS, *n* (%)	53	(28.5)	18	(27.3)	35	(29.2)	0.784
EQ-5D domain, *n* (%)							
Decreased mobility	48	(36.6)	17	(36.2)	31	(36.9)	0.933
Difficulty with self-care	54	(41.2)	26	(48.9)	31	(36.9)	0.041
Problems performing usual activities	88	(67.7)	31	(67.4)	55	(65.5)	0.825
Pain or discomfort	56	(43.8)	21	(45.7)	35	(42.7)	0.745
Anxious or depressed	51	(40.5)	18	(39.1)	33	(40.7)	0.935
EQ-5D score index, mean (SD)	0.61	(0.32)	0.62	(0.32)	0.58	(0.33)	0.549
Missing EQ-5D, *n* (%)	60	(32.3)	21	(31.8)	39	(32.5)	0.924
EQ-VAS, median (IQR)	50	(40–70)	50	(30–70)	60	(40–76.25)	0.022
Missing EQ-VAS, *n* (%)	75	(40.3)	27	(40.9)	48	(40.0)	0.904

AF—atrial fibrillation, mRS—modified Rankin Scale, EQ-5D—EuroQoL Five Dimensions, SD—standard deviation, EQ-VAS—EuroQoL visual analog scale, IQR—interquartile range. † Out of those alive at 90 days, not lost to follow-up (*n* = 133).

**Table 4 medicina-58-00800-t004:** Univariate and multivariable Cox regression analysis for overall survival of stroke patients.

Covariates		Univariate			Multivariable †		
		HR (95% CI)		*p*-Value	HR (95% CI)		*p*-Value
**Age**		1.05	(1.03–1.07)	**<0.001**	1.03	(1.01–1.06)	**0.007**
**Gender**	**Female**	1.00	(reference)				
	**Male**	0.69	(0.43–1.11)	0.125			
**Atrial fibrillation**	**No**	1.00	(reference)		1.00	(reference)	
	**Yes**	2.33	(1.44–3.75)	**<0.001**	1.27	(0.74–2.19)	0.385
**Reperfusion treatment**	**No**	1.00	(reference)				
	**Yes**	1.18	(0.73–1.90)	0.507			
**Baseline mRS ≤ 2**	**No**	1.00	(reference)		1.00	(reference)	
	**Yes**	0.33	(0.19–0.59)	**<0.001**	0.56	(0.30–1.05)	0.072
**Baseline NIHSS**		1.13	(1.10–1.17)	**<0.001**	1.11	(1.07–1.14)	**<0.001**

HR—hazard ratio, CI—confidence interval, mRS—modified Rankin Scale, NIHSS—National Institutes of Health Stroke Scale. † Akaike information criterion = 550.9.

## Data Availability

The data that support the findings of this study are available from the corresponding author upon reasonable request.

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
