# Peer review of "High Prevalence of Atrial Fibrillation in a Lithuanian Stroke Patient Cohort"

_medicina, 2022, doi:10.3390/medicina58060800_

Round 1
Reviewer 1 Report
This small (n=238 patients) but well conducted observational study evaluates the prevalence, medical support and clinical outcome of atrial fibrillation (AF) in stroke patients admitted to a leading stroke center in Eastern Lithuania. Data obtained during hospital stay were supplemented by a telephone-based follow-up 90 days post discharge. Moreover, one year case fatality was investgated. About 40% patients presented with atrial AF at admission, most of them with known preexisting AF in medical history.
Importantly, AF-patients exhibited more severe neurological impairment, and about 80% of these stroke patients with concommitant AF received insufficient anticoagulation at stroke onset. Moreover, in-hospital supervision with respect to AF was incomplete.
Although there was no significant difference in one year survival between stroke patients presenting with or without AF at hospital admission (multivariate analysis), the quality of life remained significantly impaired during a follow-up of one year.
Although the study cohort is small the presented data are of high clinical relevance, as they reflect actual clinical practice. As these data were obtained from a university reference center critical clinical data may even be more pronounced in secondary centers of rural regions.
Special comment:
- I think reader`s interest could significantly be increased if the clinical data on AF-awareness and embolic stroke prevention in different European countries are compared in a small table within the discussion section. As this topic is of high clinical relevance this may facilitate an important discussion on health care delivery in this field.
Minor comment:
- all abbreviations should be explained within an extra list
Author Response
Dear Reviewer,-
We thank you for considering our manuscript titled “High prevalence of atrial fibrillation in a Lithuanian stroke patient cohort”. Please find our replies to the comments in the attachment below.
Sincerely,
Rytis Masiliūnas
Vilnius University Faculty of Medicine
M. K. Čiurlionio str. 21, Vilnius 03101, Lithuania
Phone: +370 610 25429
Fax: +370 688 62728

Reviewer 2 Report
The authors performed a neat descriptive study on the prevalence and impacts of AF in all ischemic stroke patients in a single centre in Lithuania. The study design itself was sound, and the results were within expectations.
I only have several issues that would like the authors to address on:
- Clarify the detection and diagnosis definition of AF (at stroke onset or after). How were patients qualified as "pre-existing AF at admission"? Based on clinical history or ECG at arrival? Then, how did AF detected in the other 16 (84 minus 68) patients? Since the authors stated only 5 patients had Holter during hospitalization. Supposedly by another ECG?
- Following that, consider a study flowchart to present all the included patients (each month), and breakdown into how patients were clarified into pre-existing AF or AF detected after stroke.
- Consider compare (if possible) patients with pre-existing AF and AF detected after stroke (AFDAS). Any difference on LVO and outcome? May take a look at the reference (https://pubmed.ncbi.nlm.nih.gov/34986652/)
- Were EQ-5D all self-rated or could also be filled out by caregiver? For patients with aphasia, how were EQ-5D evaluated?
- Why were there so many patients with loss of follow-up on mRS and EQ-5D? Consider address that in the limitations.
- Consider clarify Table 2 by stating that the timing of "antithrombotic treatment status". For example for pre-existing AF, the status should be "upon admission". For all AF on discharge, the status should be "upon discharge".
- Following that, please clarify the "therapeutic range" of INR. Was it 2.5-3.5 or 2.0-3.0?
- In abstract, consider adding the statement that LVO status was only detected in those who had received vascular imaging (CTA).
- Did the authors had any data on the stroke mechanism or subtype? For example TOAST criteria. How many were cryptogenic stroke?
Author Response

(The authors gave the same response as above.)

Round 2
Reviewer 2 Report
I thank the authors for their great effort and the manuscript is largely improved now.
Only one small typo: in Figure 1, right side, if I interpreted it correctly, it should be "deceased" before discharge, n=13.
Author Response
Dear Reviewer,-
We thank you for your kind suggestions, and for pinpointing the spelling mistake in Figure 1. We have corrected it and with some additional language editing, we have uploaded the modified Manuscript with modifications highlighted in “track changes.
Thank you and we are looking forward to your final decision.
Sincerely,
Rytis Masiliūnas
Vilnius University Faculty of Medicine
M. K. Čiurlionio str. 21, Vilnius 03101, Lithuania
Phone: +370 610 25429
Fax: +370 688 62728